# Long-term observations of black carbon and carbon monoxide in the Poker Flat Research Range, central Alaska, with a focus on forest wildfire emissions

Takeshi Kinase[1], Fumikazu Taketani[1,2], Masayuki Takigawa[1], Chunmao Zhu[2], Yongwon Kim[3], Petr Mordovskoi[1], and Yugo Kanaya[1,2]

[1]Institute of Arctic Climate and Environment Research, Japan Agency for Marine-Earth Science and Technology (JAMSTEC), Yokohama 2360001, Japan
[2]Earth Surface System Research Center, Research Institute for Global Change, Japan Agency for Marine-Earth Science and Technology (JAMSTEC), Yokohama 2360001, Japan
[3]International Arctic Research Center, University of Alaska Fairbanks (UAF), Fairbanks 757340, U.S.A.

*Correspondence to*: Takeshi Kinase (tkinase@jamstec.go.jp)

**Abstract**

Forest wildfires in interior Alaska represent an important black carbon (BC) source for the Arctic and sub-Arctic. However, BC observations in interior Alaska have not been sufficient to constrain the range of existing emissions. Here, we show our observations of BC mass concentrations and carbon monoxide (CO) mixing ratios in the Poker Flat Research Range (65.12° N, 147.43° W), located in central Alaska, from April 2016 to December 2020. The medians, 10th, and 90th percentile ranges of the hourly BC mass concentration and CO mixing ratio throughout the observation period were 13, 2.9, and 56 ng m$^{-3}$ and 124.7, 98.7, and 148.3 ppb, respectively. Sporadically large peaks in the BC mass concentration and CO mixing ratio were observed at the same time, indicating influences from common sources. These BC peaks coincided with peaks at other comparative sites in Alaska, indicating large BC emissions in interior Alaska. Source estimation by FLEXPART-WRF confirmed a contribution of boreal forest wildfires in Alaska and western Canada when high BC mass concentrations were observed. For these cases, we found a positive correlation ($r = 0.44$) between the observed BC/$\Delta$CO ratio and fire radiative power (FRP) observed in Alaska and Canada. This finding implies that the variability of the BC and CO emission ratio is associated with the intensity and time progress of forest wildfires and suggests the BC emission factor and/or inventory could be potentially improved by FRP. We recommend that FRP be integrated into future bottom-up emission inventories to achieve a better understanding of the dynamics of pollutants from frequently occurring forest wildfires under the rapidly changing climate in the Arctic.

## 1 Introduction

Climate change in the Arctic region has been strongly accelerated compared to the global average (Box et al., 2019; Bonfils et al., 2020). The near-surface air temperature increased between 1.8 and 3.1 °C in the period between 1971 and 2017 (Box et al., 2019). This rapid temperature increase in the Arctic region caused decreases in the extent of sea ice (Aizawa et al., 2021), resulting in the acceleration of Arctic warming (Cohen et al., 2014; Thackeray and Hall, 2019). Even if net $CO_2$ emission is controlled to zero until the end of the 21st century (SSP1-2.6 scenario), modelling studies predicted a more than 3.5 °C temperature increase (Cai et al., 2021; Xie et al., 2022). However, there are still some difficulties associated with climate predictions based on global climate models because of the widespread use of different model hindcasts and forecasts (Overland et al., 2014). Specifically, it is known that the Arctic amplification process causes an acceleration in Arctic warming, but the process is highly complicated and is not sufficiently understood; this includes processes involved in aerosol concentration changes and the deposition of black carbon (BC) on snow and ice surfaces (Cohen et al., 2014). Thus, more research is required to understand Arctic climatic processes.

BC aerosols, which are formed by various incomplete combustion processes, such as fossil fuel and biomass burning (Bond et al., 2013), strongly contribute to warming by absorbing solar radiation (Bond et al., 2013; IPCC, 2021). In addition, BC deposited on snow and ice surfaces decreases surface albedo and contributes to snow melting and warming (Aoki et al., 2011; Bond et al., 2013; Oshima et al., 2020; IPCC, 2021). BC can be transported over long distances (estimated lifetimes are 3–6 days globally (Wang et al., 2014; Lund et al., 2018)) and affect the climate and environment of remote regions, such as the Arctic (Wang et al., 2011; Matsui et al., 2022). However, large discrepancies among model estimations for BC climate effects on the Arctic remain (Gliß et al., 2021) because of a lack of observation data (IPCC, 2021) to constrain the models in terms of dependence on emission inventories (Pan et al., 2020; Matsui et al., 2022) and/or removal rates (Ikeda et al., 2017; Lund et al., 2018). For long-range transport from Asia to the Arctic, constraints on the major BC emissions from East Asia (Choi et al., 2020; Kanaya et al., 2020), ship-based observations for BC transport to the Arctic (Taketani et al., 2016, 2022), evaluation of the multimodel bias using these datasets (Whaley et al., 2022) and an improved understanding of transport mechanisms and source attributions (Ikeda et al., 2017; Zhu et al., 2020) have been achieved. However, more observational constraints are required for the characterization of BC emissions from boreal forest wildfires (Pan et al., 2020; AMAP, 2021).

Forest wildfires in the northern American region, especially those that occur in Alaska every summer (Picotte et al., 2020), are one of the important BC emission sources in the Arctic and subarctic troposphere, and they result in depositional fluxes on snow and ice over the Arctic and surrounding regions (Xu et al., 2017; AMAP, 2021; Matsui et al., 2022). The occurrences of these forest wildfires in interior Alaska have increased since the 1980s (Sierra-Hernández et al., 2022), and this increasing trend is predicted to continue (Hu et al., 2015; Box et al., 2019; AMAP, 2021); the emission of aerosols, including BC from forest wildfires, is projected to severely affect the environment (Halofsky et al., 2020) and climate (Schmale et al., 2021) in the future.

BC mass concentrations have long been observed in the atmosphere and snow at Utqiagvik (Barrow) (Eck et al., 2009; Garrett
et al., 2011; Mori et al., 2020), which is a high Arctic coastal tundra site. Campaign studies on atmospheric BC mass
concentrations were also conducted in interior Alaska using aircrafts (Kondo et al., 2011b; Bian et al., 2013; Creamean et al.,
2018). These campaign observations have provided an in-depth understanding of aerosol parameters related to wildfires.
However, separate long-term observations of BC mass concentrations are required to characterize annual trends and seasonality.
Fewer studies have reported atmospheric BC mass concentrations in interior and coastal Alaska (Polissar et al., 1996, 1998;
Eck et al., 2009; Mouteva et al., 2015) and the high Arctic coastal site (Alert, Canada) (Garrett et al., 2011). To understand the
long-term variations in BC mass concentration and their impacts on the climate and environment, more BC observation data
from interior Alaska are needed (AMAP, 2011). In this study, we aimed to investigate detailed variations in BC mass
concentration and its sources, with a focus on forest wildfires in interior Alaska, based on our monitoring of BC and CO at the
Poker Flat Research Range (PFRR), which is a University of Alaska Fairbanks (UAF) observational site in interior Alaska.

## 74 2 Method

### 75 2.1 Observation site

We conducted BC and CO monitoring at the PFRR (65.12° N, 147.43° W, 500 m a.s.l.) starting in April 2016. The PFRR is
located in the centre of interior Alaska (Figure 1), approximately 35 km northeast of Fairbanks. The PFRR is surrounded by a
predominant evergreen needled-leaved (black spruce; *Picea mariana*) forest with shrubland and herbaceous vegetation
(Buchhorn et al., 2020). Note, that the effects of deposition by trees and canopies can be ignored because the laboratory is
located on a mountain hill, with non-tall (~2 m) sparse black spruce forest. In this study, BC and CO monitoring results were
analysed between April 2016 and December 2020.

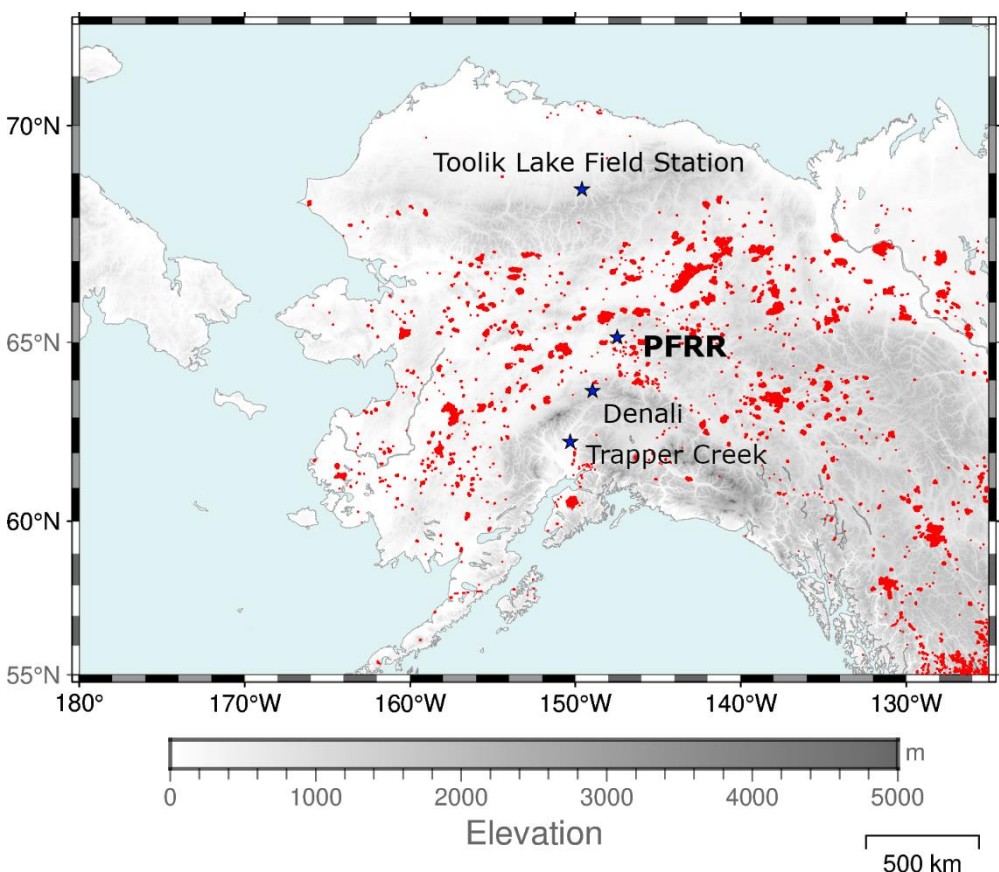


Figure 1. A map that shows the location of the PFRR and other sites compared in Section 3.2 (Trapper Creek, Denali, and Toolik Lake Field Station). All hot spots (larger than 0.3 MW in FRP) observed in the USA and Canada by VIIRS between 2016 and 2020 are shown in red colour.


## 2.2 Measurements

BC was measured by a Continuous Soot Monitoring System (BCM3130, Kanomax, Japan) with a flow rate of 0.78 L min$^{-1}$ at standard temperature and pressure (STP; 273 K and 1013 hPa). Sample air was introduced using an approximately 10 m conductive silicone tube (1/2″ i.d.) from a height of 5.5 m above the ground. The measurement technique of BCM3130 is based on filter-based optical absorption, thus other light-absorbing particles and scattering particles can be a source of interferences on BC measurement (Bond et al., 1999; Kondo et al., 2009). To minimize interferences from these particles, coarse mode particles (approximately >1.0 µm), such as mineral dust, were removed by a PM$_{1.0}$ cyclone (URG-2000-30ED,

URG, USA) operated with a small flow regulation pump (~4.5 L min$^{-1}$ at STP). Note, as most BC particles are smaller than 1
µm (Bond et al., 2013), BC loss through the PM$_{1.0}$ cyclone can be ignored. In addition, to remove nonrefractory particles, such
as sulfate and organics, the sample air was heated to approximately 300 °C using a heated inlet before it was introduced into
the instrument. More details of the instrument are described elsewhere (Miyazaki et al., 2008; Kondo et al., 2009, 2011a). One-
minute observation data were averaged to hourly data as the primary data. The limit of detection value (LOD) for hourly BC
mass concentration was estimated to be 2 ng m$^{-3}$, which is the sum of average hourly data and 3-$\sigma$ values using 18 hours of
particle-free air measurements.
The CO mixing ratio was measured by an infrared absorption photometer (48iTLE, Thermo Fisher Scientific, USA) with a
flow rate of 0.5 L min$^{-1}$. Sample air was introduced using an approximately 10 m PFA tube from a height of 5.5 m above the
ground. Internal zero measurements were carried out for 20 minutes every hour, and the CO mixing ratio was estimated from
the difference in absorption between the sample and the zero measurements. Span gas (0.99 ppm CO/N$_2$, Taiyo-Nissan, Tokyo,
Japan) calibration was performed in April 2016. We calculated $\Delta$CO as the enhancement in CO from background levels (14
days moving 5-percentile values of observation results). Cases with hourly $\Delta$CO larger than 3-$\sigma$ (13.9 ppb in median, 1-$\sigma$ was
derived from zero mode measurements before and after the hourly ambient air observations) were only used for analysis. To
validate our CO observations, we compared our observed CO mixing ratio with aircraft observations (less than 500 m AGL
above the PFRR) provided by the NOAA Global Monitoring Laboratory (https://doi.org/10.15138/39HR-9N34; accessed on
2 November 2023) (Figure S1), confirming a good agreement between these two observation results.

**2.3 Model calculation**
The FLEXPART (FLEXible PARTicle dispersion model)-WRF (Weather Research & Forecast) model was used in backward
mode to characterize the source areas and sectors for the sampled air masses at the PFRR. FLEXPART-WRF version 3.3
(Brioude et al., 2013) and WRF version 4.4 (Skamarock et al., 2019) were employed for this study. The FLEXPART-WRF
model was driven by mass-weighted wind fields and perturbation within the PBL calculated by WRF, which covers the
Northern Hemisphere with a 45-km horizontal resolution. The ERA5 global reanalysis (Hersbach et al., 2020) was used as the
initial and lateral boundary conditions of WRF, and the meteorological field of WRF was also nudged to ERA5 with e-folding
times of 3 hours and 12 hours for wind fields and temperature, respectively. Wet deposition is the major removal process for
BC, and the deposition process in FLEXPART version 10 (Grythe et al., 2017) was applied to the FLEXPART-WRF model
and was used in this study, with values of 10.0, 1.0, 0.9, and 0.1 employed as the collection efficiencies for wet deposition by
rain and snow and the activation efficiencies of cloud condensation nuclei (CCN) and ice nuclei (IN) ($C_{rain}$, $C_{snow}$, CCN$_{eff}$, and
IN$_{eff}$), respectively, which estimated by (Grythe et al., 2017) as the best parameters over several Arctic regions, i.e., Barrow,
Alert, and Zeppelin. The FLEXPART-WRF calculation was conducted every 6 hours from April 2016 to December 2020. For
each simulation, 40000 particles were released at $0.5 \times 0.5$ degrees (horizontally) and from 0 to 200 m AGL (vertically) centred
at the PFRR. The particles were tracked for 20 days at 6-hour intervals, and most simulated particles reached PFRR within
approximately 10 days (Figure 4(c)). The primary output of the FLEXPART-WRF backward calculations was the potential
emission sensitivity (PES), which expresses the residence time of particles at a given location and is used to characterize the
transport pathways of the sampled air masses. The concentration of BC was estimated by multiplying PES and emissions based
on a procedure reported by Sauvage et al. (2017). ECLIPSE (Evaluating the Climate and Air Quality Impacts of Short-Lived
Pollutants) version 6b (Klimont et al., 2017) and GFED (Global Fire Emission Database) version 4.1 (Daily) (van der Werf et
al., 2014) were used as the anthropogenic and biomass burning emissions, respectively. Note that the Chinese BC emissions
from ECLIPSE version 6b with the monthly profile of version 5 are certified with downwind atmospheric BC observations
(Kanaya et al., 2020), while other bottom-up inventories might result in a factor of ~2 overestimation. The PES fields were
calculated with a horizontal resolution of $0.5 \times 0.5$ degrees. The contribution of particles within 100 m from the surface was
considered for the calculation of PES for anthropogenic emissions. The plume height of the GFAS (Global Fire Assimilation
System) (Di Giuseppe et al., 2017) was also used for the estimation of the injection height for biomass burning emissions. The
fractional contribution of anthropogenic emissions was considered using eight sectors in the ECLIPSE emission, i.e., ship, gas
flaring, waste incineration, transport, industry, energy, domestic, and agriculture, and the anthropogenic and biomass burning
emissions were divided into eight regions, i.e., Europe, Central Asia, Russia, East Asia, Canada, Alaska, USA (excluding
Alaska), and Others. The mean age of BC was also estimated by the mean lag time between release and observed time weighted
by the amount of emission at each time period within the 20-day backward calculations.

## 2.4 Analysis of the effect of forest wildfire on the BC mass concentration at the PFRR

We characterized the observed BC/$\Delta$CO ratios, which are known to be valuable indicators of emission sources and combustion
conditions (Kondo et al., 2011b; Pan et al., 2017; Selimovic et al., 2019), in terms of fire radiative power (FRP), which accounts
for forest wildfire intensity. To do this, we compared the BC/$\Delta$CO ratio in cases of high BC mass concentrations (see section
3.4) observed between June and September (406 hours in total) and FRP observed by the Visible Infrared Imaging Radiometer
Suite (VIIRS) on the Suomi NPP satellite. Airmasses were traced for 4 days at the most using the Hybrid Single-Particle
Lagrangian Integrated Trajectory model (HYSPLIT; (Stein et al., 2015)) with GDAS1 meteorological datasets (3 h archived
$1° \times 1°$ Global Data Assimilation System) from the National Centers for Environmental Prediction
(http://ready.arl.noaa.gov/gdas1.php; accessed on 2 November 2023). The calculation started from 500 m AGL at the PFRR
site, and fire spots were searched along with the trajectories.
The BC/$\Delta$CO ratio is also affected by atmospheric processes (Kanaya et al., 2016; Choi et al., 2020), as only BC is lost via
wet removal processes. To extract observation results that were not affected by wet removal processes, we used accumulated
precipitation along the trajectory (APT) as an indicator of wet removal processes. Previous studies showed that the BC/ΔCO
ratio can be changed when APT is larger than 1 mm (Choi et al., 2020; Kanaya et al., 2016; Kondo et al., 2011b). Therefore,
the duration for the accumulation of fire spots was shortened when APT reached 1 mm or when the trajectory reached ground
level. Rectangles were defined with ±0.5° in the longitudinal direction and ±0.25° in the latitudinal direction centring around
hourly air mass positions. Then, the FRP and the number of hot spots (points) were accumulated for individual rectangles over
the duration of the trajectories. Finally, the total accumulated FRP ($\sum$FRP) was divided by the detected total points to yield an
index describing the conditions of fires affecting the observed airmasses. As hot spot datasets, VIIRS 375 m
(VNP14IMGTML_NRT) archived datasets (countries were "United States" and "Canada") from the Fire Information for
Resource Management System (FIRMS) website (https://earthdata.nasa.gov/firms; accessed on 2 November 2023) were used
in this study. The selected confidence levels were 'nominal' or 'high', and the selected type attributed to thermal anomalies
was 'presumed vegetation fire'. In addition, we used FRP values greater than 0.3 MW for each hot spot because hot spots
smaller than 0.3 MW included outliers (Figure S2). Only hot spots that were observed within the previous 24 hours were
considered. Through this procedure (hereafter, we simply use 'back trajectory'), forest wildfires in Alaska and western Canada
($\sum$FRP>0) were detected in 184 cases of hourly BC observation results. Note that we also confirmed that no back trajectories
could suggest forest wildfires in other seasons.

## 3 Results and discussion

### 3.1 Time series of observed BC and CO concentrations

The time series of BC mass concentration and CO mixing ratio are shown in Figure 2, and those of annual median, 10th, and
90th percentile values are summarized in Table 1. The median hourly BC mass concentration and 10th and 90th percentile
values throughout the observation period were 13, 3, and 56 ng m$^{-3}$, respectively. No clear increase in annual median BC mass
concentration was observed (Table 1). Observed median BC mass concentrations were the same level as previous reports at
Utqiagvik (Barrow) (12 ng m$^{-3}$), which showed BC mass concentration over the long term using the same instrument
(BCM3130) employed in this study (Sinha et al., 2017; Mori et al., 2020). Abrupt peaks (up to 5540 ng m$^{-3}$) were occasionally
observed during summer at PFRR, but these peaks were not observed at Utqiagvik. On the other hand, increases in BC mass
concentrations were reported in Utqiagvik between January and March, while not in PFRR. These different variations may be
attributed to the topological separation by the Brooks mountain range and to the polar dome structure (Quinn et al., 2007;
Sharma et al., 2013).
The median, 10th, and 90th percentiles of hourly CO mixing ratios throughout the observation period were 124.7, 99.0, and
148.2 ppb, respectively. Similar to BC, increases in the annual median CO mixing ratio were not observed, but contrary to the
BC mass concentration, the CO mixing ratio showed clear seasonal variation, high in spring (between February and April,
143.5 ppb in the median) and low in summer (July and August, 103.3 ppb in the median) (Figure 2(b)). These observed CO
mixing ratios and seasonal variations were consistent with previous studies that reported the CO mixing ratio at the PFRR
(Kasai et al., 2005; Yurganov et al., 1998). In summer, CO peaks coincident with BC mass concentration were found,
suggesting a common emission source for both BC and CO.

Table 1. Annual summary of the observed hourly BC mass concentration and CO mixing ratio at the PFRR.

| Year | BC (ng m$^{-3}$) | | | CO (ppb) | | |
| | Median | 10th percentile | 90th percentile | Median | 10th percentile | 90th percentile |
|---|---|---|---|---|---|---|
| 2016[a] | 11 | 2 | 49 | 109.7 | 93.1 | 130.3 |
| 2017 | 15 | 3 | 65 | 128.2 | 100.5 | 148.8 |
| 2018 | 14 | 3 | 53 | 118.2 | 93.3 | 149.4 |
| 2019 | 15 | 3 | 63 | 128.4 | 113.1 | 150.8 |
| 2020 | 13 | 3 | 50 | 131.3 | 107.5 | 150.6 |

[a] Observations started on 28 April 2016.

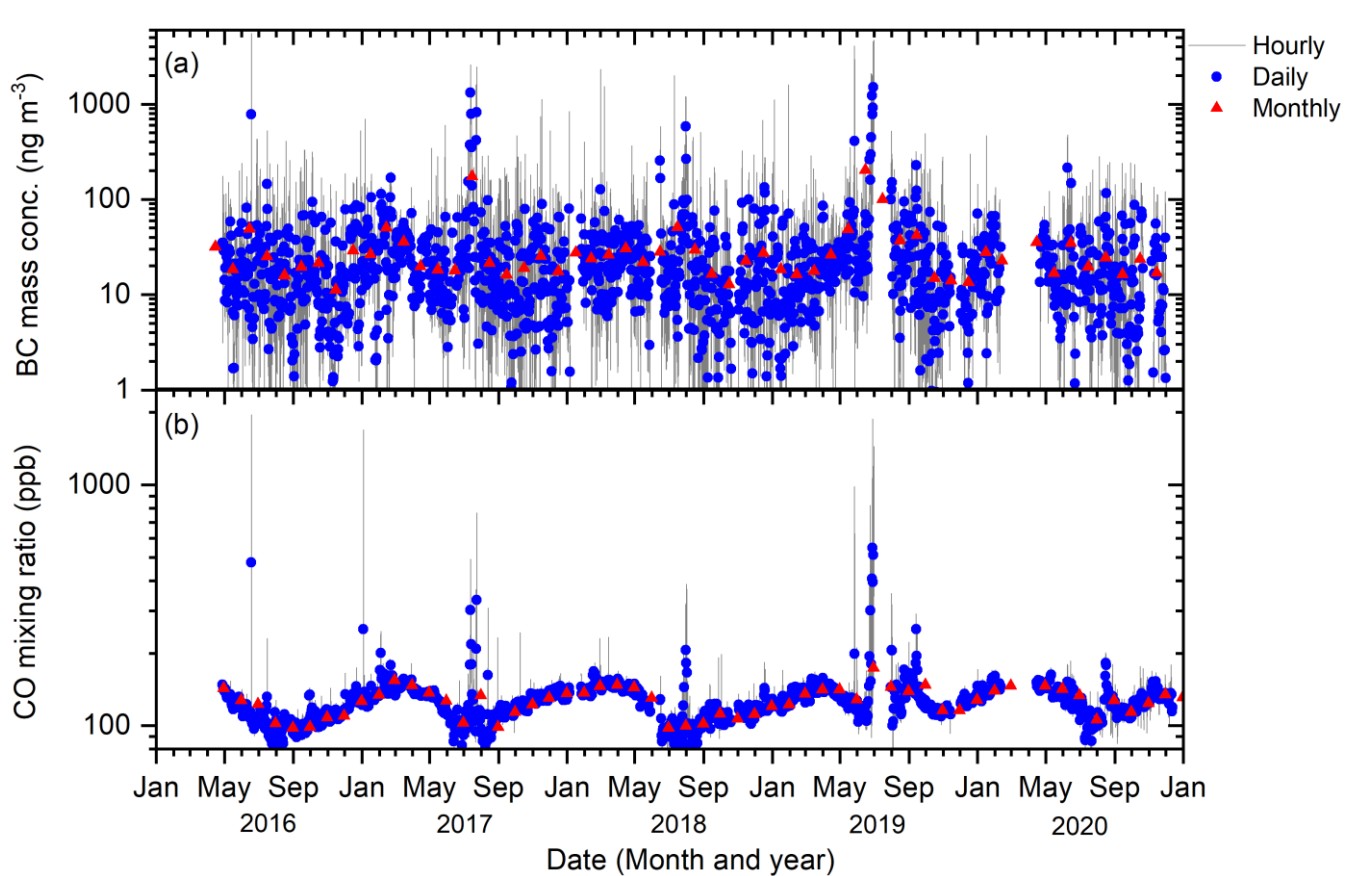


Figure 2. Time series of (a) BC mass concentration and (b) CO mixing ratio observed at the PFRR from April 2016 to December 2020. Grey lines, blue filled circles, and red filled triangles in both (a) and (b) show hourly, daily, and monthly averages, respectively.

## 3.2 Comparisons with other observation sites

We compared the BC observation results from the PFRR to those from other Alaskan sites (Table 2 and Figure S3), i.e., Trapper Creek (TRCR), Denali (DENA), and Toolik Lake Field Station (TOOL), using datasets for 24-hour filter samples collected every three days. The datasets were from the thermal/optical reflectance method at DENA, TRCR, and TOOL (http://views.cira.colostate.edu/fed/QueryWizard/; accessed on 2 November 2023). A systematic bias might be present in terms of the methods used, but it is most likely within a factor of 2 from the actual conditions based on comparisons with recent data at various sites (Miyazaki et al., 2008; Kondo et al., 2009; Kanaya et al., 2008; Kondo et al., 2011a; Ohata et al., 2021; Sinha et al., 2017). For the BC mass concentration observed at TRCR, DENA, and TOOL, datasets flagged V0 (valid value) were selected.

The BC mass concentration peaks were nearly coincided for the PFRR, DENA, TRCR, and TOOL (Figure S3). The median and maximum daily BC mass concentrations observed at each site are summarized in Table 2. The median BC mass concentrations at DENA, TRCR, and TOOL were larger than those at the PFRR by 6–19 ng m$^{-3}$ (Table 2), but the significance of the difference is unclear considering methodological differences and associated uncertainties (precision). Here, the uncertainties of the thermal/optical reflectance method varied between 12 and 14 ng m$^{-3}$ in median values during the whole observation period. Note that our BC observation, which had a better LOD (2 ng m$^{-3}$) and higher temporal resolution (1 hour), could provide more reliable data in this low range. On the other hand, the maximum BC mass concentrations were higher at the PFRR within the period with common BC peaks than at TRCR and TOOL but similar at DENA (Table 2). This indicates that strong BC emissions in central Alaska were better captured at the PFRR than at other observation sites because PFRR is the only BC-measuring site located in the central interior of Alaska and is surrounded by forest wildfire occurring regions while other BC observation sites are located on the edge or outside of interior Alaska. We will discuss source and emission ratio characterization in Sections 3.4 and 3.5 by fully utilizing the superior temporal resolution and accuracy of our observations.

Table 2. Summary of the locations of the observation sites and the BC mass concentrations in the interior Alaska.

| Site | Latitude (° N) | Longitude (° W) | Altitude (m a.s.l) | Daily BC mass concentration (ng m$^{-3}$) | |
|---|---|---|---|---|---|
| | | | | Median | Maximum |
| PFRR[a] (this study) | 65.12 | 147.43 | 500 | 18 | 920 |
| TRCR[a, b, c] | 62.32 | 150.32 | 155 | 37 | 570 |
| DENA[a, b, d] | 63.73 | 148.97 | 658 | 24 | 1044 |
| TOOL[b, e, f] | 68.64 | 149.61 | 740 | 28 | 643 |

[a] Period of data used: 28 April 2016 – 2 December 2020.
[b] Data utilised from the IMPROVE Network.
[c] TRCR: Trapper Creek.
[d] DENA: Denali.
[e] TOOL: Toolik Lake Field Station.
[f] Period of data used: 13 November 2018 – 2 December 2020.

**3.3 Comparison of observation and model simulations and possible BC sources**
Figure 3(a) shows a time series of 6-hour averages of the observation data and 6-hourly BC mass concentrations estimated by
FLEXPART-WRF simulations. FLEXPART-WRF could capture the high BC mass concentration peaks (Figure 3(a)) with a
correlation coefficient of 0.7 (Figure S4). The median of the simulated/observed ratio (observation data > LOD in this case)
was 1.0 for the whole observation period, indicating good agreement between the model simulation and observations.
The source region and source sector contributions derived from the FLEXPART-WRF simulation are shown in Figure 3(b)
and (c). The BC source sectors and regions varied clearly according to the season (Figure 3(b) and (c)). In the warm season
(between May and September), the possible BC source regions were Russia (3.6–74 % in the 10–90 percentile) and Alaska
(12–85 % in the 10–90 percentile) and sometimes Canada (1.0–21 % in the 10–90 percentile) (Figure 3(b)), and the possible
source sector was estimated to be biomass burning (8.1–88 % in the 10–90 percentile) (Figure 3(c)), especially when BC mass
concentration was high, suggesting that BC contributions from biomass burning that occurred in Russia, Alaska, and Canada
are dominant for BC mass concentrations at the PFRR. As snow cover disappears from the ground and the atmospheric
conditions become drier, forest wildfires caused by lightning increase in these warm seasons (Reap, 1991; Kaplan and Lau,
2021), resulting in increases in BC emissions from biomass burning (AMAP, 2021). We will focus on these cases of high BC
mass concentrations from Alaska and discuss the relationship between forest wildfire intensity and the BC/ΔCO ratio in the
following section.
On the other hand, in the cold seasons (between October and April), the domestic (24–48 % in the 10–90 percentile) and
transport sectors (25–48 % in the 10–90 percentile) were estimated to be possible dominant BC source sectors (Figure 3(c)).
The dominant source region was Alaska (19–88 % in the 10–90 percentile), and occasionally, Russia (0.89–31 % in the 10–
90 percentile) and East Asia (1.2–41 % in the 10–90 percentile) contributed to the BC mass concentration in PFRR (Figure
3(b)).

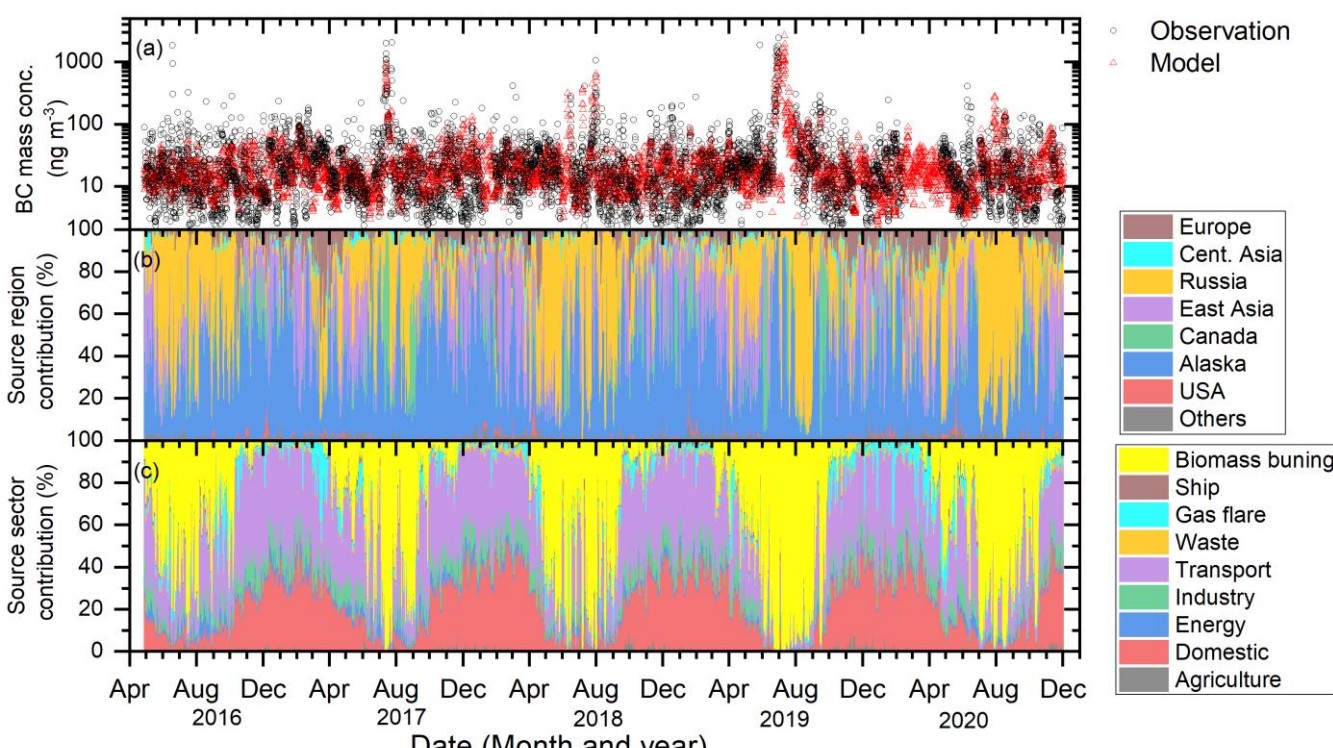


Figure 3. Time series of (a) BC mass concentrations, (b) attribution of BC at the PFRR to source regions, and (c) to source
sectors. Black open circles and red open triangles in (a) show the 6-hour average observations and 6-hourly simulations,
respectively. Individual colour bars in (b) and (c) depict the estimated contributions from the source regions and sectors,
respectively. The FLEXPART-WRF model was used for all simulations.

**3.4 Biomass burning contribution for cases of high BC mass concentrations**
Hereafter, we focus on cases of high BC mass concentrations at the PFRR (647 hours in total), which were selected with the
98 percentile value (171 ng m$^{-3}$) as the threshold for the hourly BC mass concentration. The cumulative BC mass concentration

observed in these cases of high BC mass concentrations accounted for 5.7–43 % of the annual BC mass concentration, although the duration of these periods was very short (17–187 hours in a year). Most of these cases of high BC mass concentrations (approximately 90 %) were observed in warm seasons (between June and September) and were related to forest wildfires in Alaska. The median CO mixing ratio for the cases of high BC mass concentrations (174.7 ppb) was also higher than that in other periods (124.7 ppb), suggesting that both BC and CO were emitted from forest wildfires (see Section 3.3).

The normalized frequency distribution of the BC/$\Delta$CO ratio for the cases of high BC mass concentrations is shown in Figure 4(a). The median, 10th, and 90th percentile values of the BC/$\Delta$CO ratio during these periods were 4.7, 1.8, and 18 ng m$^{-3}$ ppb$^{-1}$, respectively. These observed BC/$\Delta$CO ratios in the cases of high BC mass concentrations were in the same range or sometimes higher than those in previous studies that reported the BC/$\Delta$CO ratios from boreal forest wildfire emissions in Canada (Kondo et al., 2011b) and Siberia (Paris et al., 2009; Chi et al., 2013; Vasileva et al., 2017).

The medians of the fractional contributions of biomass burning on total BC mass concentrations and the mean age of BC estimated by the FLEXPART-WRF simulation in these cases of high BC mass concentrations (>154 ng m$^{-3}$ in 6-hourly average) were higher and shorter (95.5 % and 2.6 days) than those in other periods (7.6 % and 6.9 days) (Figure 4(b) and (c)), indicating a strong contribution of BC from neighbouring forest wildfires (Figure S5). We also calculated the 6-hourly mass-weighted biomass burning contributions from individual source regions (6 categories based on Figure 3(b), Central Asia and Europe are included in Others) to the BC mass concentrations at the PFRR (Figure 5). As a result, we found that large peaks, such as those observed between June and August in 2017, 2018, and 2019, coincided well with the peaks of BC contributions mostly from forest wildfires in Alaska (Figure S5). BC from forest wildfires that occurred in western Canada also affected the BC concentration at PFRR (Figure S6) but to a lesser frequency. Russia was also estimated as an effective BC source region (Figure 3), but BC concentration did not exceed 0.1 µg m$^{-3}$ in most cases (Figure 5). These results confirmed that the observed cases of high BC mass concentrations were primarily affected by local forest wildfires in Alaska. These peaks were widely observed in Alaska (Section 3.2) and imply a large impact of local forest wildfires on BC mass concentration in this region. However, when these cases of high BC mass concentrations were selected, the median of the simulated/observed ratio was 0.30, indicating underestimation in the model simulation (possibly due to insufficient spatial resolution for neighbouring forest wildfires and difficulties in representing the vertical profiles of BC emissions) or/and in emission inventories in the cases of high BC mass concentrations. Several studies have indicated that differences in different inventories cause large uncertainties in model estimates of BC emissions, atmospheric concentrations, and radiative impacts, especially in boreal North America (Carter et al., 2020; Pan et al., 2020). The impact of different inventories on model estimates will be discussed in the future.

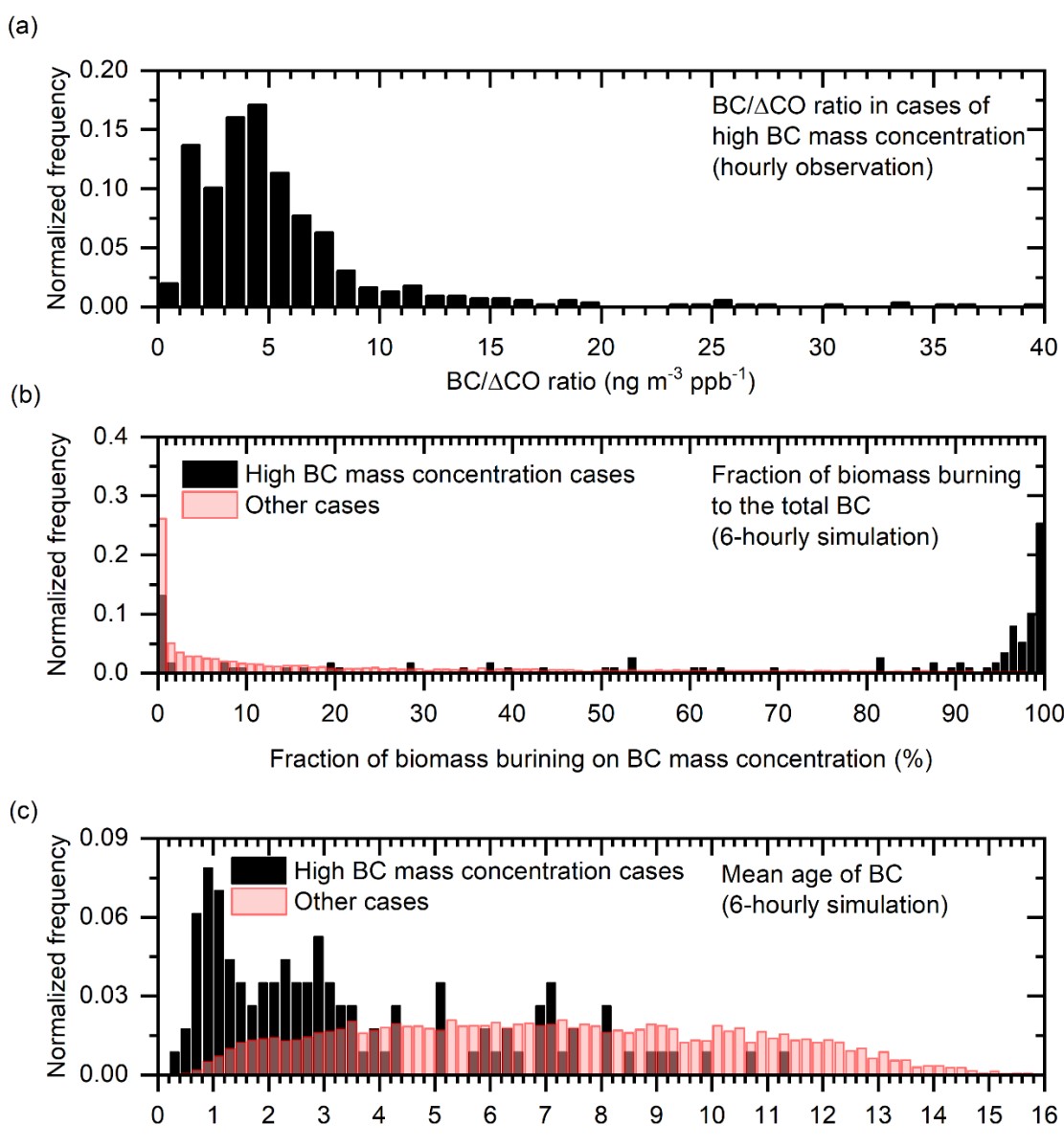

Figure 4. (a) Histogram of the observed hourly BC/ΔCO ratio at the PFRR in cases of high BC mass concentrations (>98 percentile). Histograms of the simulated 6-hourly (b) fractions of BC mass concentrations from biomass burning to the total BC and (c) mean age of BC estimated by the FLEXPART-WRF model. Black and red bars in (b) and (c) show the cases of high BC mass concentrations and the other cases (<98 percentile), respectively.


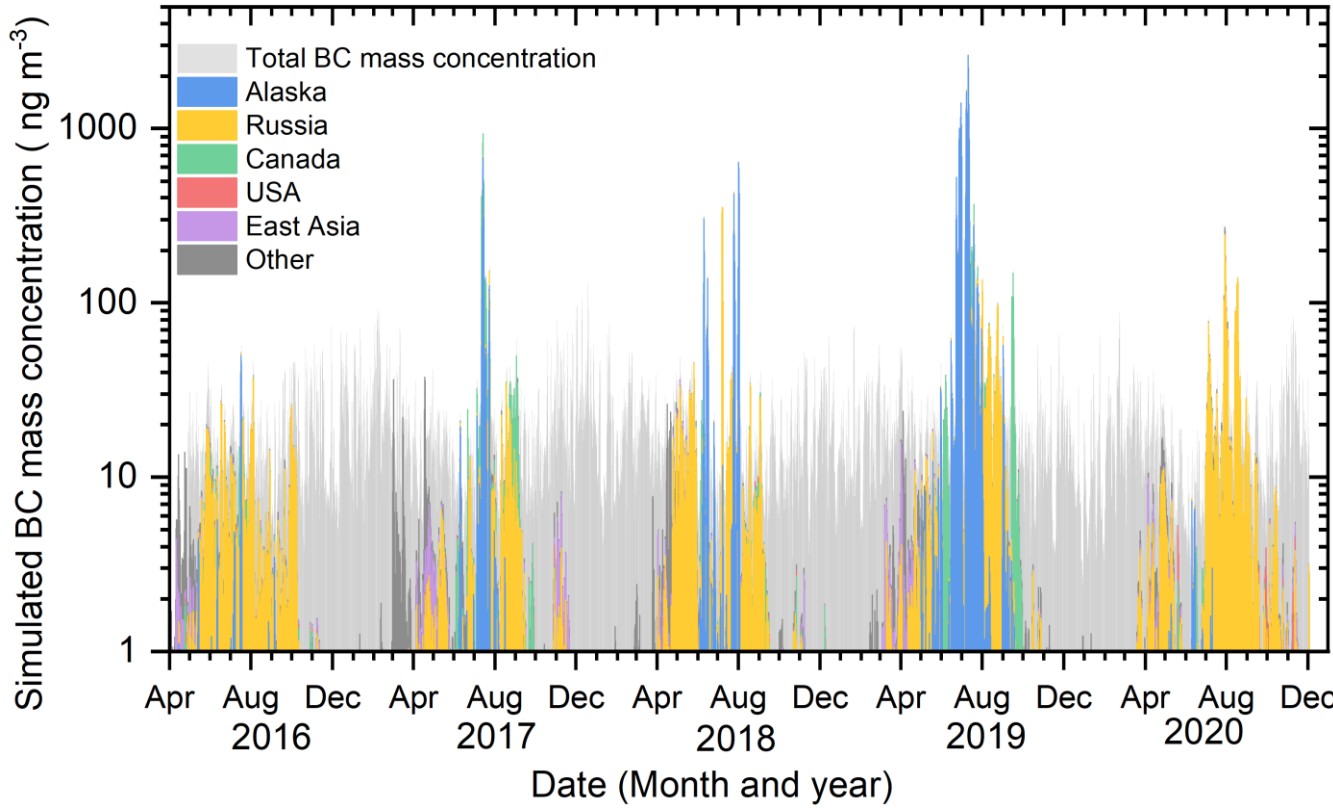


Figure 5. A time series of the 6-hourly BC mass concentrations at the PFRR simulated by the FLEXPART-WRF model. Light grey bars show the total BC mass concentrations. Other individual colour bars (overlaid on the light grey bars) show the BC mass concentrations for biomass burning from each source region.



**3.5 Relationship between the BC/ΔCO ratio and FRP**

In the previous section, we showed that most cases of high BC mass concentrations were related to forest wildfires in Alaska. Increases in biomass-burning-derived BC/ΔCO ratios with combustion efficiency were suggested from an observational study on boreal forest wildfire (Kondo et al., 2011b) and from laboratory-scale burning experiments of crop residues (Pan et al., 2017); however, in-depth studies examining variabilities in BC/ΔCO ratios based on long-term, near-forest observations have not been conducted. To consider the possibility that combustion conditions (flaming and smouldering) primarily control the

BC/ΔCO ratio, we are going to investigate the relationship between the BC/ΔCO ratio and forest wildfire intensity in this
section.
We found a positive correlation ($r$ = 0.44, p < 0.0001, n = 184) between the BC/ΔCO ratio and ∑FRP/point values (Figure 6).
This positive correlation between the BC/ΔCO ratio and $\Sigma$FRP/point values, represented for the first time to our knowledge,
is qualitatively consistent with previous studies that showed that high combustion efficiency (larger than 0.9 in modified
combustion efficiency value (MCE)) increased BC/ΔCO ratios (Selimovic et al., 2019; Kondo et al., 2011b; Pan et al., 2017),
which is related to the fact that the BC production process is mostly related to the flaming process (high MCE), while that of
CO is related to the smouldering process (low MCE). For example, Pan et al. (2017) measured BC, CO, and $CO_2$ from biomass
burning in small-scale combustion experiments. In their experiment, dry and wet wheat straw samples and dry rapeseed plant
samples were burned, and the time evolution of BC/ΔCO ratio and MCE were observed. They reported that BC is mostly
produced during the flaming process, and the evolution of the BC/ΔCO ratio which depends on the combustion stage could be
confirmed (13.9±10.1 ng m$^{-3}$ ppbv$^{-1}$ for MCE larger than 0.95 cases, and less than 7.1 ng m$^{-3}$ ppbv$^{-1}$ for MCE smaller than
0.96 cases). Although these BC/ΔCO ratios are larger than our observed BC/ΔCO ratio, differences in fuels might be a possible
reason. Selimovic et al. (2018) also burned some types of fuels, including coniferous trees, in a large indoor combustion facility
and measured BC, CO, and $CO_2$ with various other chemical species. They reported a high BC/ΔCO ratio (13.8 ng m$^{-3}$ ppbv$^{-1}$
on average) and a low BC/ΔCO ratio (4.7 ng m$^{-3}$ ppbv$^{-1}$ on average) in the condition of flaming-dominated and smouldering-
dominated, respectively, in the same range as our observed values. Moreover, Chakrabarty et al. (2016) tested Alaskan peat
and Siberian peat in the combustion chamber under smouldering conditions, and low BC/ΔCO ratios (1.2–2.6 ng m$^{-3}$ ppbv$^{-1}$)
were reported. The positive relationship between the BC/ΔCO ratio and MCE is also observed in the field measurements
(Kondo et al., 2011b; Selimovic et al., 2019). Although MCE and FRP are different parameters, both parameters indicate
combustion conditions and have a strong correlation (Wiggins et al., 2020). Therefore, for the first time, we report a positive
correlation between the BC/ΔCO ratio and FRP as a combustion condition indicator. The wide range of BC/ΔCO ratios reported
from boreal forest wildfires, from 1.7–3.4 ng m$^{-3}$ ppb$^{-1}$ (Kondo et al., 2011b) to 6.1–6.3 ng m$^{-3}$ ppb$^{-1}$ (Vasileva et al., 2017),
could be better explained when the index introduced here ($\Sigma$FRP/point) is considered. This relationship should be taken into
account when constructing future emission inventories from boreal forest wildfires.
A positive correlation was found after optimizing the spatial window size (±0.5° in the longitudinal direction and ±0.25° in
the latitudinal direction), in which hot spots were taken into account for each hour along the trajectory (from -96 to 0 hours),
and the associated time window was used to determine coincident fires that affected the observations (from -24 to 0 hours).
Based on the spatial resolution of GDAS1 (1° × 1°), we set our initial windows as ±0.5° for latitude and longitude. However,
PFRR is in a high latitude and the geometrical length of latitude is approximately 2 times longer than that of longitude. For
this reason, we defined latitudinal width as ±0.25° finally. Although we tested finer window size cases, a similar positive trend

was confirmed. The remaining scatter might have arisen from differences in the detection of hot spots in the presence of clouds (Li et al., 2018). To overcome shortcomings in hot spot detection, improvements in the frequency of hot spot scanning should be made, for example, via the use of MODerate Resolution Imaging Spectroradiometer (MODIS) combined with VIIRS observations; it should be noted, however, that there is a bias in FRP observations between MODIS and VIIRS, especially for boreal forests (Li et al., 2018). Improvements in the accuracy and consistency of FRP analysis between multiple satellite observations can facilitate a more in-depth understanding of the relationship between FRP and the BC/$\Delta$CO ratio.

The simulated/observed ratios in cases of high BC mass concentrations were low (0.30, Section 3.3), contrary to the good agreement observed in overall cases (1.0, Section 3.3). The BC/$\Delta$CO ratios in commonly used emission inventories are 4.9 ng m$^{-3}$ ppb$^{-1}$ for GFED4s (van der Werf et al., 2017) and 4.4 ng m$^{-3}$ ppb$^{-1}$ for Andreae (2019) and are in a range similar to that of our median BC/$\Delta$CO ratio. However, our observed BC/$\Delta$CO ratios in cases of high BC mass concentrations for forest wildfires had a broad range between 1.7 and 7.3 ng m$^{-3}$ ppb$^{-1}$ at the 10 and 90 percentiles (median was 4.2 ng m$^{-3}$ ppb$^{-1}$), respectively, related to the $\sum$FRP/point values. This implies that the BC emission factors from biomass burning could vary depending on the FRP. Although several previous inventory studies used FRP for the estimation of activity data (Carter et al., 2020), namely, fuel burned or burned area, no inventories included the evolution of the emission factors of BC and/or CO. Our findings suggest the potential for improving BC emission inventories and/or emission factors by using FRP. In addition, BC emission estimation using satellites would be improved by using our results. CO emissions estimated by satellite observations are sometimes used to estimate other pollutant emissions from forest fires using emission ratios derived from in situ measurements (Zheng et al., 2023). As its extension, BC emissions could be estimated, regarding our quantified BC/$\Delta$CO ratios and their evolutions with FRP directly as the emission ratio of BC to CO. The frequency of boreal forest fires may increase in the future (Box et al., 2019; Hu et al., 2015); as a result, their impact on climate and air quality might become more severe in Alaska and the Arctic (Kim et al., 2005; Schmale et al., 2018; Stohl et al., 2006). Our long-term observations of BC and CO at an hourly temporal resolution in the interior of Alaska provide unique information to test model simulations and emission inventories relevant to the climate and air quality of the Arctic.

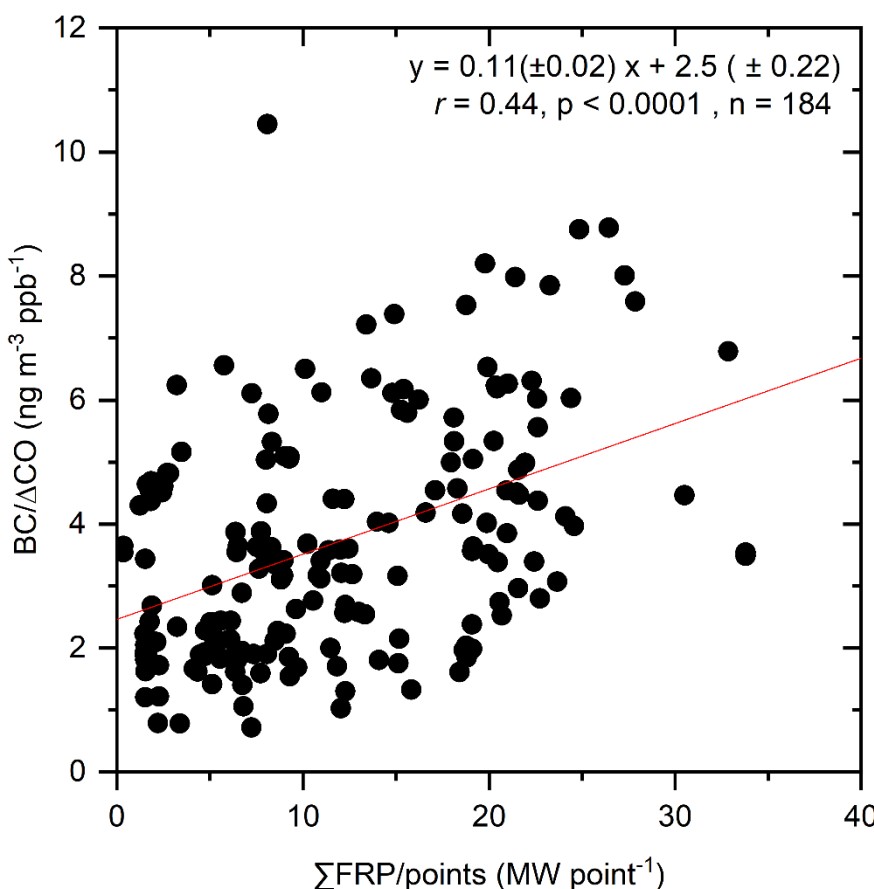


Figure 6. The scatter plot (black filled circles) between the hourly BC/ΔCO ratio observed at the PFRR and the ∑FRP/point
values in cases of high BC concentrations (>98 percentile). Data from June to September were analyzed. The ∑FRP/point
values are the average FRP of the hot spots (points) present along the backward trajectories from the PFRR (see section 2.4).
A red line indicates the result of a linear regression fit.
**4 Conclusion**
We showed key features of the BC and CO concentrations observed at the PFRR in interior Alaska since 2016 in this paper.
The annual medians of the BC mass concentration and CO mixing ratio were 11–15 ng m$^{-3}$ and 109.7–131.3 ppb, respectively.
Large and short-term increases in BC mass concentrations were sometimes observed between June and September. A clear
seasonal variation was observed in the CO mixing ratio, which was high in spring (between February and April, 143.5 ppb in
the median) and low in summer (July and August, 103.3 ppb in the median). The CO mixing ratio coincided with the high BC
mass concentration peaks, suggesting a strong contribution from forest wildfires to BC and CO concentrations.
The BC mass concentrations observed at other sites in Alaska, i.e., DENA, TRCR, and TOOL, were compared with our results.
The annual median BC mass concentrations at the PFRR were lower than those at TRCR, DENA, and TOOL, but coinciding
BC mass concentration peaks were found at these observation sites. In these cases of high BC mass concentrations, BC mass
concentrations at the PFRR were larger than those at TRCR and TOOL but similar at DENA, indicating that strong BC
emissions from forest wildfires occurred in interior Alaska and affected broad areas in Alaska.
The dominance of forest wildfires in Alaska as a major cause of high BC mass concentration was also supported by the model
simulations. We simulated BC mass concentration using the FLEXPART-WRF model and compared the simulations with the
observation results. The model simulation could capture observational results ($r = 0.70$) in which the median
simulated/observed ratio was 1.0. The estimated BC source sectors and regions were biomass burning from Russia, Alaska,
and sometimes Canada between May and September, while those for other periods were domestic sources and transport and
were mainly from Alaska.
When we focused on cases of high BC mass concentrations (greater than 98 percentile values), we found that forest wildfires
occurring in Alaska were the dominant source of BC in those cases from the model simulation results. The mean ages of BC
and biomass burning contributions in these cases of high BC mass concentrations estimated by FLEXPART-WRF were 2.6
days and 95.5 %, respectively, relatively shorter and higher than those in other cases (6.9 days and 7.6 %, respectively). The
peaks of the calculated biomass burning contributions from Alaska to BC mass concentrations at the PFRR coincided well
with observed and simulated peaks in cases of high BC mass concentrations, suggesting that the forest wildfires that occurred
around the PFRR are important.
The median observed BC/$\Delta$CO ratio in cases of high BC mass concentrations related to forest wildfires was 4.2 ng m$^{-3}$ ppb$^{-1}$
and was in the same range as that in previous studies reporting the BC/$\Delta$CO ratio of boreal forest wildfire emissions. Finally,
we tracked airmass origin for 4 days using the HYSPLIT model with FRP satellite observations in these cases and investigated
the relationship between the observed BC/$\Delta$CO ratio and FRP, which was normalized by the number of hot spots (points)
observed by VIIRS. A positive correlation was found between these parameters ($r = 0.44$). For the first time, the properties of
the BC/$\Delta$CO ratio from boreal forest wildfires were systematically characterized in terms of FRP, suggesting the potential to
improve emission inventories and/or emission factors by using FRP.

**Data availability**
The BC and CO observation results are available online (https://ads.nipr.ac.jp/dataset/A20241101-003). We used public data
for BC observation results at Denali, Trapper Creek, and Toolik Lake Field Station
(http://views.cira.colostate.edu/fed/QueryWizard/).

## Supplement

The supplement related to this article is available online at https://doi.org/xxxxxxxx.

## Author contributions

TK, FT, CZ, YKi, and YKa conducted and recorded observations for BC and CO at the PFRR site. MT conducted the FLEXPART-WRF model simulations. YKi assisted in the fieldwork at the PFRR site. TK, FT, MP, and YKa summarized the observation results, and TK wrote the first draft with MT. All authors contributed to the discussion and writing of the manuscript.

## Competing interests

At least one of the (co-)authors is a member of the editorial board of Atmospheric Chemistry and Physics.

## Acknowledgement

The authors acknowledge technical support from Dr. Takuma Miyakawa, a researcher at JAMSTEC and help with field work from Dr. Hideki Kobayashi, a researcher at JAMSTEC. The authors also thank all the supporting members at JAMSTEC. The authors thank NOAA ARL for providing the CO aircraft observation data, HYSPLIT model, and GDAS1 meteorological data. We also thank the IMPROVE network. IMPROVE is a collaborative association of state, tribal, and federal agencies and international partners. The US Environmental Protection Agency is the primary funding source, with contracting and research support from the National Park Service. The Air Quality Group at the University of California, Davis, was the central analytical laboratory, and carbon analysis was carried out by the Desert Research Institute. We also thank the anonymous reviewers for their precise and valuable comments that greatly improved the paper.

## Financial support

This work was funded by the Arctic Challenge for Sustainability II (ArCS II), Program Grant Number JPMXD1420318865, the Arctic Challenge for Sustainability (ArCS), Program Grant Number JPMXD1300000000, and a National Research Foundation of Korea Grant from the Korean Government (MSIT; the Ministry of Science and ICT, NRF-2021M1A5A1065425) (KOPRI-PN24011).

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
