# Peer review of "Long-term observations of black carbon and carbon monoxide in the"

_EGUsphere, 2023_

## Author Comment (AC1)

Black text: Community comments by Dr. Jian Liu

Blue text: Our replies

Red text: Modified parts of the manuscript

The authors have conducted a lot of analysis work in this paper, demonstrating a new way to help people understand the emissions over Arctic. There are some issues :

→We appreciate the comments by Dr. Jian Liu. All the comments were helpful in improving our manuscript. Please see our answers to the specific comments below.

1) There are several models mentioned in this paper, as we know, models are impacted from the model itself, parameters, and input, such as emissions, so how did you find a balance between the models and their uncertainties, any assumptions included? If yes, more details should be given in the supplement.

→ Thank you for pointing this out. Though we used Hysplit in the analysis where FRP was related to the air mass transport, the model we mainly used in this manuscript was FLEXPART-WRF. The model was well validated and used for the estimation of source-receptor analysis (Liu et al., 2015; Zhu et al., 2020; Raut et al., 2017). More detailed information for the FLEXPART-WRF calculation in this study is added as Table S1 in the supplement file. Although the errors associated with the model's transport and wet removal may become larger with long-range transport, the uncertainties with emissions would outweigh for the cases affected by biomass burning, which is the main subject of this manuscript. We have added a summary table of RMSEs when using 6 different types of biomass burning emission inventories in Table S2.

"Table S1. Main configuration parameters adopted for the FLEXPART-WRF simulations in this study.

| WRF-ARW 4.2.1 configuration | |
| --- | --- |
| Initial and boundary conditions | ECMWF ERA5 (Pressure-levels), hourly |
| PBL parameterization | MYNN level 2.5 |
| Shortwave and longwave radiation | RRTMG |
| Land surface | Noah-MP |
| Microphysicis | Morrison 2-moment |
| Convection | Grell 3D ensemble |
| Maximum height | 10 hPa |
| Domain size (x,y and z) | 399x399x44 grids (mass points) |
| Map projection | Polar Stereographic |

centered at the North Pole

| FLEXPART-WRF 3.3 configuration | |
|---|---|
| LSUBGRID | 1 (enables subgrid terrain effect) |
| TURB_OPTION | 1 (PBL turbulent mixing is calculated in the same manner with FLEXPART) |
| CBL_OPTION | 1 (skewed option for the convective PBL) |
| SFC_OPTION | 1 (PBL height is taken from WRF) |
| WIND_OPTION | 1 (mass-weighted, time-averaged wind U,V and W calculated in WRF) |
| Interval of input data | hourly |
| Density of BC | 1400 kg m$^{-3}$ |
| Mean diameter of BC | 0.25 μm |
| Standard deviation of BC size distribution | 1.25 |
| Release height | 100-200 m A.G.L. |
| Number of particles | 40,000 |
| Output grids (horizontal) | 180°W-180°E, 20°-90°N, every 0.5° |
| Output grids (vertical) | 11 layers (50, 100, 150, 200, 500, 1000, 1500, 2000, 2500, 3000, 20000 m A.G.L.) |

"

2) FRP has widely been utilized in the top-down emission inventories, but for bottom-up emission researchers, the burned area and emission factors are used to estimate the biomass burning emission. Of course, the FRP should have some correlations with wildfires and these are acknowledged as two different technique routes for emission researchers. So the question here is how to help these researchers use the FRP or help each other, the authors should give accurate details in this part. If not, the conclusion will be weak. The authors should also make it clear that this might be useful to the bottom-up emissions in the results section.

→ Given that the emission rate is estimated as a product of activity data (AD) and emission factor (EF), previous inventory studies used FRP for the estimation of AD, namely, fuel burned or burned area. On the other hand, our study suggested that FRP information is also essential in improving EF estimation, in that the EFs of BC and/or CO dynamically changes with FRP. Previous studies generally only crudely assume the BC emission factors are constant over different combustion conditions. Though needing a total carbon (or CO2, as its major fraction) emission information for a full parameterization, we here proposed a new principle that the BC emission factor expression can be improved by taking combustion conditions (related to FRP) into account. This is clearly different from previous studies.

In addition, BC emission estimation using satellites would be improved by using our results. CO emissions estimated by satellite observations are sometimes used to estimate other pollutant emissions from forest fires using emission ratios derived from in situ measurements (Zheng et al., 2023). As its extension, BC emissions could be estimated, regarding our quantified BC/ΔCO ratios and their evolutions with FRP directly as the emission ratio of BC to CO.

We will modify our manuscript accordingly after we receive other reviewer's comments.

3) Following Question 2), based on this recently published paper (https://doi.org/10.5194/acp-24-367-2024), why did the authors choose GFED only as the emission inventory in the FLEXPART-WRF, how about using an FRP-based emission inventory and checking if there was some relation between BC/△CO and the differences for two different routes' emission inventories?

→ Thank you for pointing out an important paper. As indicated by this comment, some inventories have already used FRP to estimate AD, but they have not used to modify EF. Although we compared the model calculation results using the footprint calculated by the FLEXPART-WRF and 6 different emission inventories (FINNv1.5 and v2.5, GFED, GFAS, QFED, and FEER), including those using FRP (GFAS, QFED, and FEER), we found that GFED showed better agreement with the concentration observed at PFRR comparing to other inventories. This model comparison will be presented in detail in a separate paper (in prep.). For this reason, we showed only the GFED result in this paper. It would

be ideal that FRP is used for estimating both AD and EF, as this study suggested; however, development of such an emission inventory and simulations using it will require more work and is out of scope in this paper. This suggestion is important and will be considered in the future. We added a summary table (Table S2) for tested emission inventories and their performance in the supplement file. "Table S2. Root Mean Square Error (RMSE) of the 6-hour averaged model calculated BC mass concentrations at PFRR. Model-calculated BC mass concentrations were estimated using the same footprint calculated by the FLEXPART-WRF model and different emission inventories.

| Inventory | RMSE (ng m$^{-3}$) |
|---|---|
| FINN v1.5 (Wiedinmyer et al., 2011)) | 35.8 |
| FINN v2.5 (MODIS+VIIRS) (Wiedinmyer et al., 2023) | 28.4 |
| GFED v4.1s (Randerson et al., 2015) | **12.7** |
| GFAS v1.2 (Kaiser et al., 2012) | 26.6 |
| QFED v2.5r1 (https://portal.nccs.nasa.gov/datashare/ iesa/aerosol/emissions/QFED/v2.5r1/) | 21.9 |
| FEER v1.0-G1.2 (Ichoku and Ellison, 2014) | 31.0 |

"

4) In Fig. S2, there are some very high values from ground-based observations but not shown by NOAA hourly-average observations, does that mean the authors has a different data curation method from NOAA, if yes, why?

→ Because the NOAA aircraft observations were done approximately every 3 weeks, short peaks in between the opportunities were easily missed. We added a sentence below to the caption of Figure S2 and modified Figure S2.

"

[Figure]

Figure S2. Time series of CO mixing ratios observed at PFRR. Red circles show the hourly ground-based observation (this study), and black points show the averages of individual aircraft observations (below 500 m a.g.l) made by NOAA Global Monitoring Laboratory approximately every 3 weeks. Because our observations were done continuously with a high-temporal resolution, several high-concentration peaks could be captured.
"

5) Some minor revisions should be corrected, e.g. Line 81, "section 3.2" should be "Section 3.2" and so on. Lines 405, and 416, the references should include the DOI link, and other similar issues in the reference part.

→ Thank you for your kind indication. We corrected Line 81, Lines 405, 406, and other lines. In addition, related to the comment 2, we added three references (Akagi et al., 2011; Wiggins et al., 2021; Zheng et al., 2023).

[Modified and added references]

"Akagi, S. K., Yokelson, R. J., Wiedinmyer, C., Alvarado, M. J., Reid, J. S., Karl, T., Crounse, J. D.,

and Wennberg, P. O.: Emission factors for open and domestic biomass burning for use in atmospheric models, Atmos. Chem. Phys., 11, 4039–4072, https://doi.org/10.5194/acp-11-4039-2011, 2011."

[revised manuscript text omitted]

(References used in this reply)

Akagi, S. K., Yokelson, R. J., Wiedinmyer, C., Alvarado, M. J., Reid, J. S., Karl, T., Crounse, J. D., and Wennberg, P. O.: Emission factors for open and domestic biomass burning for use in atmospheric models, Atmos. Chem. Phys., 11, 4039–4072, https://doi.org/10.5194/acp-11-4039-2011, 2011.

Ichoku, C. and Ellison, L.: Global top-down smoke-aerosol emissions estimation using satellite fire radiative power measurements, Atmos. Chem. Phys., 14, 6643–6667, https://doi.org/10.5194/acp-14-6643-2014, 2014.

Kaiser, J. W., Heil, A., Andreae, M. O., Benedetti, A., Chubarova, N., Jones, L., Morcrette, J.-J., Razinger, M., Schultz, M. G., Suttie, M., and van der Werf, G. R.: Biomass burning emissions estimated with a global fire assimilation system based on observed fire radiative power, Biogeosciences, 9, 527–554, https://doi.org/10.5194/bg-9-527-2012, 2012.

Liu, D., Quennehen, B., Darbyshire, E., Allan, J. D., Williams, P. I., Taylor, J. W., Bauguitte, S. J.-B., Flynn, M. J., Lowe, D., Gallagher, M. W., Bower, K. N., Choularton, T. W., and Coe, H.: The importance of Asia as a source of black carbon to the European Arctic during springtime 2013, Atmos.

Chem. Phys., 15, 11537–11555, https://doi.org/10.5194/acp-15-11537-2015, 2015.

Randerson, J. T., Van Der Werf, G. R., Giglio, L., Collatz, G. J., and Kasibhatla, P. S.: Global Fire Emissions Database, Version 4.1 (GFEDv4), https://doi.org/10.3334/ORNLDAAC/1293, 2015.

Raut, J.-C., Marelle, L., Fast, J. D., Thomas, J. L., Weinzierl, B., Law, K. S., Berg, L. K., Roiger, A., Easter, R. C., Heimerl, K., Onishi, T., Delanoë, J., and Schlager, H.: Cross-polar transport and scavenging of Siberian aerosols containing black carbon during the 2012 ACCESS summer campaign, Atmos. Chem. Phys., 17, 10969–10995, https://doi.org/10.5194/acp-17-10969-2017, 2017.

Wiedinmyer, C., Akagi, S. K., Yokelson, R. J., Emmons, L. K., Al-Saadi, J. A., Orlando, J. J., and Soja, A. J.: The Fire INventory from NCAR (FINN): a high resolution global model to estimate the emissions from open burning, Geoscientific Model Development, 4, 625–641, https://doi.org/10.5194/gmd-4-625-2011, 2011.

Wiedinmyer, C., Kimura, Y., McDonald-Buller, E. C., Emmons, L. K., Buchholz, R. R., Tang, W., Seto, K., Joseph, M. B., Barsanti, K. C., Carlton, A. G., and Yokelson, R.: The Fire Inventory from NCAR version 2.5: an updated global fire emissions model for climate and chemistry applications, Geoscientific Model Development, 16, 3873–3891, https://doi.org/10.5194/gmd-16-3873-2023, 2023.

Wiggins, E. B., Andrews, A., Sweeney, C., Miller, J. B., Miller, C. E., Veraverbeke, S., Commane, R., Wofsy, S., Henderson, J. M., and Randerson, J. T.: Boreal forest fire CO and $CH_4$ emission factors derived from tower observations in Alaska during the extreme fire season of 2015, Atmos. Chem. Phys., 21, 8557–8574, https://doi.org/10.5194/acp-21-8557-2021, 2021.

Zheng, B., Ciais, P., Chevallier, F., Yang, H., Canadell, J. G., Chen, Y., van der Velde, I. R., Aben, I., Chuvieco, E., Davis, S. J., Deeter, M., Hong, C., Kong, Y., Li, H., Li, H., Lin, X., He, K., and Zhang, Q.: Record-high CO2 emissions from boreal fires in 2021, Science, 379, 912–917, https://doi.org/10.1126/science.ade0805, 2023.

Zhu, C., Kanaya, Y., Takigawa, M., and Ikeda, K.: FLEXPART v10. 1 simulation of source contributions to Arctic black carbon, Atmospheric, 2020.

---

## Author Comment (AC2)

Black text: Referee comments by Anonymous Referee #1

Blue text: Our replies

Red text: Modified parts of the manuscript

* Line number is the number in the tracked manuscript.

The manuscript by Kinase et al. (Long-term observations of black carbon and carbon monoxide in the Poker Flat Research Range, central Alaska, with a focus on forest wildfire emissions) reports ~ 4 years of continuous monitoring data of black carbon (BC) and carbon monoxide (CO). The observation data were simulated by the FLEXPART-WRF model. The data were analyzed by focusing on high-concentration events that were likely induced by biomass burning. The authors suggest that the emission ratio of BC may depend on fire radiation power (FRP). The topic is within the scope of the journal. The organization and writing quality of the manuscript can be improved. I suggest all the authors of the manuscript carefully read through it again to improve both the general organization and expressions of individual contents. I have the following comments.

 →Thank you for your kind review and important comments. All the comments were helpful for us in improving our manuscript. Please see our answers to the specific comments below.

Major comment

I believe that Figure 6 shows the most important message of the manuscript. Although there might be a relationship between BC/delta_CO and FRP, the data are highly scattered (r = 0.44). At the current moment, it is very difficult for me to evaluate if such a trend exists. Comparison with some other data may help better understand the data. For instance, BC/delta_CO data from laboratory experiments (smoldering, flaming) can be compared with the present dataset. If the existing dataset from the laboratory supports the idea, the conclusion can be strengthened.

 →Thank you for your important comment. There are several studies that have tested the relationship between the BC/$\Delta$CO ratio and the modified combustion efficiency value (MCE) to characterise the BC/$\Delta$CO ratio as a function of combustion intensity. For example, Pan et al. (2017) measured BC, CO, and $CO_2$ in small-scale combustion experiments. In their experiment, dry and wet wheat straw samples and dry rapeseed plant

samples were burned, and the time evolution of the BC/$\Delta$CO ratio and MCE were observed. They reported that BC is mostly produced during the flaming process, and the evolution of the BC/$\Delta$CO ratio which depends on the combustion stage could be confirmed (13.9$\pm$10.1 ng m$^{-3}$ ppbv$^{-1}$ for MCE larger than 0.95 cases, and less than 7.1 ng m$^{-3}$ ppbv$^{-1}$ for MCE smaller than 0.96 cases). Although these BC/$\Delta$CO ratios are larger than our observed BC/$\Delta$CO ratio, differences in fuels might be a possible reason. Selimovic et al. (2018) also burned some types of fuels, including coniferous trees, in a large indoor combustion facility and measured BC, CO, and CO$_2$ with various other chemical species. They reported a high BC/$\Delta$CO ratio (13.8 ng m$^{-3}$ ppbv$^{-1}$ on average) and a low BC/$\Delta$CO ratio (4.7 ng m$^{-3}$ ppbv$^{-1}$ on average) in the conditions of flaming dominated and smouldering dominated, respectively, in the same range as our observed values. Moreover, (Chakrabarty et al., 2016) tested Alaskan peat and Siberian peat in the combustion chamber under smouldering conditions, and low BC/$\Delta$CO ratios (1.2–2.6 ng m$^{-3}$ ppbv$^{-1}$) were reported.

The positive relationship between the BC/$\Delta$CO ratio and MCE is also observed in the field measurements (Kondo et al., 2011b; Selimovic et al., 2019). Although MCE and FRP are different parameters, both parameters indicate combustion conditions and have a strong correlation (Wiggins et al., 2020). Therefore, our observed evolution of BC/$\Delta$CO ratios with increasing FRP can be a reasonable result. We modified our manuscripts as shown below.

"lines 328–340: For example, Pan et al. (2017) measured BC, CO, and CO$_2$ from biomass burning in small-scale combustion experiments. In their experiment, dry and wet wheat straw samples and dry rapeseed plant samples were burned, and the time evolution of BC/$\Delta$CO ratio and MCE were observed. They reported that BC is mostly produced during the flaming process, and the evolution of the BC/$\Delta$CO ratio which depends on the combustion stage could be confirmed (13.9$\pm$10.1 ng m$^{-3}$ ppbv$^{-1}$ for MCE larger than 0.95 cases, and less than 7.1 ng m$^{-3}$ ppbv$^{-1}$ for MCE smaller than 0.96 cases). Although these BC/$\Delta$CO ratios are larger than our observed BC/$\Delta$CO ratio, differences in fuels might be a possible reason. Selimovic et al. (2018) also burned some types of fuels, including coniferous trees, in a large indoor combustion facility and measured BC, CO, and CO2 with various other chemical species. They reported a high BC/$\Delta$CO ratio (13.8 ng m$^{-3}$ ppbv$^{-1}$ on average) and a low BC/$\Delta$CO ratio (4.7 ng m$^{-3}$ ppbv$^{-1}$ on average) in the condition of flaming-dominated and smouldering-dominated, respectively, in the same range as our observed values. Moreover, Chakrabarty et al. (2016) tested Alaskan peat and Siberian peat in the combustion chamber uunder a smouldering conditions, and low BC/$\Delta$CO ratios (1.2–2.6 ng m$^{-3}$ ppbv$^{-1}$) were reported. The positive relationship between the BC/$\Delta$CO ratio and

MCE is also observed in the field measurements (Kondo et al., 2011b; Selimovic et al., 2019).…"

Other comments

Line 17

The observation period needs to be clearly stated, including the end of the observation.

→Modified.

"line 17: from April 2016 to December 2020."

Line 18

It is better to show some statistics (e.g., 10th and 90th tile values), rather than only showing the median.

→Modified

"lines 17–19: The medians, 10th, and 90th percentile ranges of the hourly BC mass concentration and CO mixing ratio throughout the observation period were 13, 2.9, and 56 ng m$^{-3}$ and 124.7, 98.7, and 148.3 ppb, respectively."

Introduction

There are too many usages of 'significant.' Reserve the usage of such an adjective only for really meaningful cases (e.g., demonstrating statistical significance).

→We checked 'significant' throughout our manuscript and changed the wording or deleted where necessary.

Line 88

The reviewer is not familiar with the observation site. Based on the description in section 2.1, it seems that the observation site is located in a forest. Can the influence of forest

canopy on particle deposition be ignored when the sampling height is 5.5 m above the ground level?

→Observatory is located on a mountain hill, with non-tall (~2m) sparse black spruce forest. Therefore, deposition effects can be ignored. We added a sentence shown below.

" Line 79–80: Note, that the effects of deposition by trees and canopies can be ignored because the laboratory is located on a mountain hill, with non-tall (~2m) sparse black spruce forest."

Line 89

Are there any reasons why the authors used the $PM_{1.0}$ cyclone? Most BC data are collected using $PM_{2.5}$ inlets. The usage of $PM_{1.0}$ cyclone makes comparison with literature data to be difficult.

→As pointed out by this comment, the $PM_{2.5}$ cyclone has been used in many studies. However, also the $PM_{1.0}$ cyclone has been use in as many other BC studies (Kondo et al., 2011a; Mori et al., 2020; Selimovic et al., 2018; Vakkari et al., 2018). The mass median diameter and count median diameter of BC particles are between 120nm and 160nm and between 50nm and 80 nm, respectively, most of BC particles are submicron particles (Bond et al., 2013). Therefore, particle loss through the $PM_{1.0}$ cyclone can be ignored for BC concentration measurement and our result can be compared with previous studies that used a $PM_{2.5}$ inlet. Coarse particles are known to interfere with the BC measurement by filter-based optical absorption technique (Bond et al., 1999). To minimize the interferences by these particles and to achieve high accuracy of BC measurement, it is better to use $PM_{1.0}$ cyclones. We added a sentence showing below.

"lines 95–96: Note, as most BC particles are smaller than 1 µm, BC loss through the $PM_{1.0}$ cyclone can be ignored."

line 113-115

How well are these parameters constrained by previous studies? It would be helpful to have some references so that readers can find the original papers for these numbers.

→Parameters for the BC removal process used in this study were chosen from (Grythe et al., 2017), which conducted a sensitivity test for the combination of BC removal parameters

for BC removal over Barrow, Alert, and Zeppelin (cf. Table 4 of their paper). We have conducted similar sensitivity tests with our observations and found that this combination showed the lowest RMSE with the observations. We added an explanation below.

"lines 124–125: which estimated by Grythe et al. (2017) as the best parameters over several Arctic regions, i.e., Barrow, Alert, and Zeppelin"

Line 130

What kind of back trajectory model was employed? Are back trajectory calculations for ground surface trustable for the duration of 20 days?

→FLEXPART-WRF version 3.3 (Brioude et al., 2013) and WRF version 4.4 (Skamarock et al., 2019) were employed in this research. The meteorological field near the surface was recalculated using WRF with increased vertical layers near the surface (around 14 layers within 0–2km from the surface) from ERA5 pressure-level data (around 9 layers in 1000–800hPa), and the vertical mixing within the PBL was diagnosed using the meteorological field of WRF. For the advection, hourly-averaged mass-weighted winds, which were calculated in WRF, were used to conserve the total mass. Although we conducted 20-day backward calculations, most simulated particles reached PFRR within approximately 10 days (Figure 4(c)). We therefore believe that our simulation is trustworthy. We added an explanation below.

"lines 127–128: , and most simulated particles reached PFRR within approximately 10 days (Figure 4(c))"

Line 166

Concentrations of most aerosol species are different from one place to another place. The description is not very informative. It would be better to describe 1) how the concentration level of the present data is compared with previous studies, and 2) what kind of geographical differences might have induced the difference.

→Thank you for your important suggestion. The median hourly BC concentration at PFRR was 13 ng m$^{-3}$ and that at Utqiagvik was 12 ng m$^{-3}$, almost the same level. Relatively high BC concentrations were observed at Utqiagvik between January and March, however, BC concentrations at PFRR did not increase in the same seasons. This difference may be attributed to the BC accumulation in the polar dome (Quinn et al., 2007; Sharma et al.,

2013). On the other hand, large peaks of BC concentration (up to 5540 ng m$^{-3}$) were sometimes observed at PFRR, however, these peaks were not observed at Utqiagvik. This difference is possibly caused by the topological separation by the Brooks mountain range. We modified our manuscript as below.

"lines 174–183: Observed median BC mass concentrations were the same level as previous reports at Utqiagvik (Barrow) (12 ng m$^{-3}$), which showed BC mass concentration over the long term using the same instrument (BCM3130) employed in this study (Sinha et al., 2017; Mori et al., 2020). Abrupt peaks (up to 5540 ng m$^{-3}$) were occasionally observed during summer at PFRR, but these peaks were not observed at Utqiagvik. On the other hand, increases in BC mass concentrations were reported in Utqiagvik between January and March, while not in PFRR. These different variations may be attributed to the topological separation by the Brooks mountain range and to the polar dome structure (Quinn et al., 2007; Sharma et al., 2013)."

Line 172

The comparison of the CO data should belong to the method section.

→We moved comparisons with aircraft observations provided by NOAA to the method section to support the accuracy of our CO observations. A part of the comparison with previous studies remained in the result section.

(moved)

" line 108–111: To validate our CO observations, we compared our observed CO mixing ratio with aircraft observations (less than 500 m AGL above the PFRR) provided by the NOAA Global Monitoring Laboratory (https://doi.org/10.15138/39HR-9N34; accessed on 2 November 2023) (Figure S1), confirming a good agreement between these two observation results."

Line 186

Figure S3 is not well organized. It may be better to separate the panel.

→Modified.

"

[Figure]

Figure S3. …To show the variation of BC mass concentrations in the low concentration level, an expanded plot for low BC mass concentrations ranging between 0 and 200 ng m-3 was shown in (b) by a linear scale."

Line 202

Why was the observation suitable for obtaining the data that were influenced by wildfires?

→Because PFRR is the only site located in the central interior of Alaska while other BC observation sites are located on the edge or outside of interior Alaska. Most forest wildfires

occur in the interior of Alaska (Figure 1). Therefore, PFRR is surrounded by forest wildfire occurring regions, differently from the other BC observation sites (Figure 1), resulting in higher BC concentration peaks than other sites. We modified our manuscript as below.

"lines 220–222: because PFRR is the only BC-measuring site located in the central interior of Alaska and is surrounded by forest wildfire occurring regions while other BC observation sites are located on the edge or outside of interior Alaska."

Line 254

Are there any typical criteria to separate smoldering and flaming by FRP?

→No typical criteria. As we explained in the reply to the major comment, there are some studies that investigated the relationship between the BC/ΔCO ratio and MCE, however, the relationship between the BC/ΔCO ratio and FRP has never been introduced. Therefore, our result (Figure 6) for the evolution of the BC/ΔCO ratio with increases in FRP will be the first report.

 L290

Back trajectories only suggest the origin of air masses. It does not indicate anything about the occurrence of wildfires.

→We analyzed as if the observed airmass was affected or not by the forest wildfire by using back trajectory analysis coupled with the FRP satellite observation (section 2.4). So we added 'with FRP (hereafter, we simply use 'back trajectory')' to Line 315.

"lines 318–321: We selected 406 hourly cases between June and September from the data selected in Section 3.4 as high BC cases from forest wildfires and chose 184 cases of hourly BC observations results affected by near forest wildfires detected in Alaska and western Canada by back trajectory analysis with FRP (hereafter, we simply use 'back trajectory')"

"lines 407: Finally, we tracked airmass origin for 4 days using the HYSPLIT model with FRP satellite observations…

L298

I am not sure if the correlation can be considered as being robust.

→We deleted 'robust' and 'clear' from this sentence.

"line 341: … , we report a positive correlation…"

"line 344: This relationship should be taken…"

L303

The selection of the window is somewhat arbitrary. How does the original data (without classification by the window) look like? Why do the authors think that such a window is needed? These points will need to be clearly described.

→Generally, airmass location estimated by the trajectory analysis can not be pinpointed, having some spreads. The original VIIRS observation has a very fine resolution (375 m), and this resolution will be too fine to detect the actual areas of forest fire. Therefore, we defined an FRP detection window for this analysis.

We used GDAS1 meteorological data for back trajectory analysis which has a 1° × 1° spatial resolution. Based on this constraint, we set our initial windows as ±0.5° for latitude and longitude. However, PFRR is in a high latitude and the geometrical length of latitude is approximately 2 times longer than that of longitude. For this reason, we defined latitudinal width as ±0.25° finally. This will take into account fires within an area of ~25 × 25 km$^2$. Although we tested some finer spatial and time resolution cases, a similar positive trend was considered, a similar positive trend was confirmed. We modified our manuscript as below.

[revised manuscript text omitted]

---

## Author Comment (AC3)

Black text: Referee comments by Anonymous Referee #2

Blue text: Our replies

Red text: Modified parts of the manuscript

* Line number is the number in the tracked manuscript.

Kinase et al., 2023 "Long-term observations of black carbon and carbon monoxide in the Poker Flat Research Range, central Alaska, with a focus on forest wildfire emissions" presents long-term observations of BC mass concentrations and CO mixing ratios in the Poker Flat Research Range, which is located in Central Alaska. The authors use the FLEXPART-WRF model to estimate contributions from black carbon source regions and sectors and the HYSPLIT model to trace airmasses that originate from forest wildfires. This study shows a good correlation between the BC/ΔCO ratio and the fire radiative power (FRP) and concludes that the FRP should be used to improve emission inventories. This work provides very useful datasets and novel ideas. The methods and analysis support well the conclusion of this study. The text is well-written although in some parts the structure needs to be carefully revised. I have some minor comments for the authors to consider, presented below:

→Thank you for your kind review and important comments. All the comments were helpful for us in improving our manuscript. Please see our answers to the specific comments below.

Abstract: Although the measurements of this study were taken in Alaska, the high BC mass concentrations were originated from wildfires in both Alaska and Western Canada (section 3.5). Therefore, I would suggest discussing in the abstract about wildfires at high latitudes rather than just in interior Alaska. I would also suggest to mention already in the abstract the whole period of sampling at the PFRR.

→Thank you for your important comment. We extended the source regions from only Alaska to boreal forests in Alaska and Canada. We also added a whole period of observations as below.

"line 17: from April 2016 to December 2020."

"line 22: a contribution of boreal forest wildfires in Alaska and western Canada"

"lines 395–396: The estimated BC source sectors and regions were biomass burning from Russia, Alaska, and sometimes Canada…"

Section 2.2: Some additional information about COSMOS is needed here. At least it should be added that this is a filter-based absorption technique.

→Modified.

"lines 91–93: The measurement technique of BCM3130 is based on filter-based optical absorption, thus other light-absorbing particles and scattering particles can be a source of interferences on BC measurement (Bond et al., 1999; Kondo et al., 2009)."

Lines 163-164: Unclear sentence; please, rephrase.

→Thank you for your important comment. We discussed for the differences in BC concentration at PFRR and Utqiagvik again and modified to be more specific. Please check.

"lines 174–183: The median hourly BC mass concentration and 10th and 90th percentile values throughout the observation period were 13, 3, and 56 ng m$^{-3}$, respectively. No clear increase in annual median BC mass concentration was observed (Table 1). Observed median BC mass concentrations were the same level as previous reports at Utqiagvik (Barrow) (12 ng m$^{-3}$), which showed BC mass concentration over the long term using the same instrument (BCM3130) employed in this study (Sinha et al., 2017; Mori et al., 2020). Abrupt peaks (up to 5540 ng m$^{-3}$) were occasionally observed during summer at PFRR, but these peaks were not observed at Utqiagvik. On the other hand, increases in BC mass concentrations were reported in Utqiagvik between January and March, while not in PFRR. These different variations may be attributed to the topological separation by the Brooks mountain range and to the polar dome structure (Quinn et al., 2007; Sharma et al., 2013)."

Lines 201-202: Why are the black carbon emissions better captured in central Alaska? Is Alaska the only source region of the measured black carbon? A discussion on the topography of the sites and meteorological conditions is needed to support this statement.

→As shown in Figure 1, most boreal forest wildfires occurred in the interior of Alaska and western Canada, and PFRR is located in the centre of these forest wildfire-occurring areas, i.e., PFRR is surrounded by strong BC emission sources, while other observation sites compared in this study are located on the edge or outside of forest wildfire occurring regions.

As reviewer #2 mentioned and we showed in Section 3.3, other boreal forests, i.e., Russia and Canada, can also affect the BC concentration at PFRR. However, as shown in Figure 5, most large peaks (exceeding 1 µg m$^{-3}$) were associated with Alaskan forest wildfires. Also, BC from the Russian forest wildfires can affect BC concentration at PFRR, however, BC concentration did not exceed 0.1 µg m$^{-3}$ in these cases, likely due to the deposition and diffusion during their long-range transport. BC from Canadian forest wildfires also affected PFRR BC concentration, however, it was a minor case. We added the sentence below.

"lines 220–222: because PFRR is the only BC-measuring site located in the central interior of Alaska and is surrounded by forest wildfire occurring regions while other BC observation sites are located on the edge or outside of interior Alaska."

"lines 285–287: BC from forest wildfires that occurred in western Canada also affected the BC concentration at PFRR (Figure S6), but to a lesser frequency. Russia was also estimated as an effective BC source region (Figure 3), but BC concentration did not exceed 0.1 µg m$^{-3}$ in most cases (Figure 5)."

Lines 219-223: Figure 2 shows also that emissions in Canada contribute to BC concentrations at the PFRR during the warm season. How important is this contribution? I would suggest discussing this in this section since you will link later the high BC concentrations with the forest wildfires in Canada.

→Fractions of BC from Canada in warm seasons were 1.0–21% in the 10–90 percentile, relatively lower than that of Alaska and Russia. However, as pointed out by this comment, sometimes BC plumes from western Canada were captured at PFRR during BC high concentration periods. Therefore, we added this information to the manuscript as shown below.

"line 241: and sometimes Canada (1.0–21% in the 10–90 percentile)"

"lines 285–286: BC from forest wildfires that occurred in western Canada also affected the BC concentration at PFRR (Figure S6) but to a lesser frequency.

"

Lines 251-256: I would suggest moving this text somewhere in the next section because it is not relevant to section 3.4.

 →Modified. These sentences were moved to Lines 312–318.

"lines 312–318: In the previous section, we showed that most high BC mass concentration cases were related to forest wildfires in Alaska. Increases in biomass-burning-derived BC/ΔCO ratios with combustion efficiency were suggested from an observational study on boreal forest wildfires (Kondo et al., 2011b) and from laboratory-scale burning experiments of crop residues (Pan et al., 2017); however, in-depth studies examining variabilities in BC/ΔCO ratios based on long-term, near-forest observations have not been conducted. To consider the possibility that combustion conditions (flaming and smouldering) primarily control the BC/ΔCO ratio, we are going to investigate the relationship between the BC/ΔCO ratio and forest wildfire intensity in this section. We selected …"

Line 259: Can you support the statement here with the back trajectory analysis?

 →We added example cases of footprints when Alaskan and Canadian forest wildfires affected the BC concentration at PFRR in Figures S5 and S6 in our supporting information.

"

Examples of PES footprints and contributions of BC from biomass burning emissions at 12:00 on 27 June 2019.

[Figure]

Figure S5. Examples of (a) a PES footprint on a global scale, (b) that focused on Alaska and Western Canada, (c) a BC emission contribution from biomass burning on a global scale, and (d) that focused on Alaska and Western Canada at 12:00 on 27 June 2019. In this case, high BC concentration (1611 ng m$^{-3}$ on hourly average) was observed at PFRR and most BC came from forest wildfires around PFRR. Star marks indicate the PFRR location.

Examples of PES footprints and contributions of BC from biomass burning emissions at 00:00 on 12 July 2017.

[Figure]

Figure S6. Same as Figure S5 but at 00:00 on 12 July 2017. In this case, high BC concentration (204 ng m$^{-3}$ on hourly average) was observed at PFRR and most BC came from forest wildfires in western Canada and partly Alaska. Star marks indicate the PFRR location.

lines 278–280: The medians of the contributions of biomass burning and the mean age of BC estimated by the FLEXPART-WRF simulation in these high BC mass concentration cases were higher and shorter (95.5% and 2.6 days) than those in other periods (7.6% and 6.9 days) (Figure 4(b) and (c)), indicating a strong contribution of BC from neighbouring forest wildfires (Figures S5)."

Line 261: East Asia is a separate category in Fig. 5. The text should be corrected.

→Our explanation in the sentence was wrong, the text in Figure 5 is correct. We modified our explanation in the manuscript.

"line 281–282: 6 categories based on Figure 3(b), Central Asia and Europe are included in Others"

Figure 4b. Typo in the label (x-axis).

→Modified.

“

[Figure]

Figure 5: I would suggest the x-axis to be the same as in Figure 2 for easier comparison.

→Modified. In addition to this comment, the colour profile has been changed to the same as in Figure 2.

“

"

Lines 291-292: Unclear sentence; please, rephrase.

→There was too much information in that sentence and it probably made it confusing and unclear. Therefore, we deleted redundant parts as below.

"lines 322–323: We found a positive correlation ($r = 0.44$, $p < 0.0001$, $n = 184$) between the BC/$\Delta$CO ratio and $\sum$FRP/point values (Figure 6)."

Line 311: There is a typo.

→Modified.

"line 357: BC/$\Delta$CO"

---

## Author Response (AR2)

**Reply to the editor's comments**

Black text: editor's comments

Red text: our replies and modified parts

**Public justification (visible to the public if the article is accepted and published)**:

Dear authors,

thank you for your revisions and responses to the reports by the reviewers. These have now also undergone another assessment by a reviewer. Based on this and my own assessment I accept your manuscript subject to minor revisions. Please address the issues below in a revised version of your manuscript.

Reply:

Thank you for your valuable feedback and decisions. We have modified our manuscript accordingly and made our observation data publicly available online. Consequentry, we have modified the data availability statement of our data to "The BC and CO observation results at the PFRR site are available online (https://ads.nipr.ac.jp/dataset/A20241101-003)." in L431-432. Please review these modifications.

Kind regards,

Philip Stier

1) Much of your analysis is correlation based. However, correlations on their own do not imply causation. Statements such as in your abstract " we found a positive correlation (r = 0.44) between the observed BC/ΔCO ratio and fire radiative power (FRP) observed in Alaska and Canada. This finding indicates that the BC and CO emission ratio is controlled by" are therefore too strong and should be reworded so that they do not automatically imply causation from correlation.

Reply:

Thank you for your indication. We have modified our sentence as below. Please check.

L24-25: "This finding implies that the variability of the BC and CO emission ratio is associated with the intensity and time progress of forest wildfires…"

L352: reworded to "implies"

2) Given that the analysis is correlation based, "significant" should only be used for well defined statistical significance. You have partially addressed this issue in response to a reviewer comment but there remain ambiguous occurrences that should be carefully reworded.

Reply:

We checked again the parts which used "significant" and removed or reworded. Please check.

L33: removed

L158: removed

L238: reworded to "dominant"

L246: reworded to "contributed to the BC mass concentration in PFRR"

L261: removed

L344: removed

3) Your methodological description of how you select trajectories affected by wildfires is not detailed enough to be reproducible (lines 319-321). This needs to be fully reproducible from your description and should be moved from the results to the methods section.

Reply:

Thank you for your comment and suggestion. The data selection for the trajectory analysis was based on data used in section 3.4, specifically focusing on measurements conducted between June and September. Trajectories were calculated for all these selected periods, resulting in 184 cases where trajectories detected forest fires in Alaska and Canada ($\sum FRP>0$). We have moved this description to the method section as below. Please check.

L148-149: "we compared the BC/$\Delta$CO ratio in high BC mass concentration cases (see section 3.4) observed between June and September (406 hours in total)..."

L164: "archived datasets (countries were "United States" and "Canada") from..."

L167: "we used FRP values greater than 0.3 MW for each hot spot"

L169-171: "Through this procedure (hereafter, we simply use 'back trajectory'), forest wildfires in Alaska and western Canada ($\sum FRP>0$) were detected in 184 cases of hourly BC observation results. Note that we also confirmed that no back trajectories could suggest forest wildfires in other seasons."

---

## Author Response (AR3)

**Reply to the editor's comments**
Black text: editor's comments
Red text: our replies and modified parts

Public justification (visible to the public if the article is accepted and published):

Dear authors,

thank you for considering the remaining outstanding issues in your revised manuscript, which I have now accepted for publication subject to technical corrections.
Reply:
We, the authors, would like to thank the editor for his cooperation and judgement. We checked and corrected our manuscript accordingly. Please check if our collections are in line with the points.

Required corrections:
- Please make all figure and table caption self explanatory. They should cleary indicate the data displayed or reproduced and fully understandable without having to find the corresponding section in the main text.
Reply:
We modified our manuscript as suggested by the editor. Additionally, we corrected four points in the sentence as shown below. Please check.

In manuscript:
L150: "VIIR" changed to "VIIRS" .
L161: "hot spots" changed to "hot spots (points)".
L162: "spot number" changed to "points".
L346: "abias" changed to "a bias".

Additionally, with the caption corrections, "high BC mass concentration cases" in the text has been changed to "cases of high BC mass concentrations."

Captions:
Figure 1. A map that shows the location of the PFRR and other sites compared in Section 3.2 (Trapper Creek, Denali, and Toolik Lake Field Station). All hot spots

(larger than 0.3 MW in FRP) observed in the USA and Canada by VIIRS between 2016 and 2020 are shown in red colour.

Figure 2. Time series of (a) BC mass concentration and (b) CO mixing ratio observed at the PFRR from April 2016 to December 2020. Grey lines, blue filled circles, and red filled triangles in both (a) and (b) show hourly, daily, and monthly averages, respectively.

Table 2. Summary of locations for observation sites and the BC mass concentrations in the interior Alaska.
(footnotes)
[a] Data were selected for the periods that all locations are available.
[b] Observations started on 13 November 2018.
[c] TRCR; Trapper Creek.
[d] DENA; Denali.
[e] TOOL; Toolik Lake Field Station.
[f] Period of data used: 13 November 2018 – 2 December 2020.

Figure 3. Time series of (a) BC mass concentrations, (b) attribution of BC at the PFRR to source regions, and (c) to source sectors. Black open circles and red open triangles in (a) show the 6-hour average observations and 6-hourly simulations, respectively. Individual colour bars in (b) and (c) depict the estimated contributions from the source regions and sectors, respectively. The FLEXPART-WRF model was used for all simulations.

Figure 4. (a) Histogram of the observed hourly BC/ΔCO ratio at the PFRR in cases of high BC mass concentrations (>98 percentile). Histograms of the simulated 6-hourly (b) fractions of BC mass concentrations from biomass burning to the total BC and (c) mean age of BC estimated by the FLEXPART-WRF model. Black and red bars in (b) and (c) show the cases of high BC mass concentrations and the other cases (<98 percentile), respectively.

Figure 5. A time series of the 6-hourly BC mass concentrations at the PFRR simulated by the FLEXPART-WRF model. Light grey bars show the total BC mass concentrations. Other individual colour bars (overlaid on the light grey bars) show the BC mass concentrations for biomass burning from each source region.

Figure 6. The scatter plot (black filled circles) between the hourly BC/ΔCO ratio observed at the PFRR and the ∑FRP/point values in cases of high BC concentrations (>98 percentile). Data from June to September were analyzed. The ∑FRP/point values are the average FRP of the hot spots (points) present along the backward trajectories from the PFRR (see section 2.4). A red line indicates the result of a linear regression fit.